# Sleep problems were unrelated to social media use in the late COVID-19 pandemic phase: A cross-national study

**Tore Bonsaksen** [1,2] *, **Daicia Price** [3], **Gary Lamph** [4,5], **Isaac Kabelenga** [6], **Amy Østertun Geirdal** [7]

**1** Faculty of Social and Health Sciences, Department of Health and Nursing Sciences, University of Inland Norway, Elverum, Norway, **2** Fcaulty of Health Sciences, Department of Health, VID Specialized University, Stavanger, Norway, **3** Department of Social Work, University of Michigan, Ann Arbor, United States of America, **4** School of Nursing and Midwifery, Keele University, Keele, Staffordshire, United Kingdom, **5** Midlands Partnership University NHS Foundation Trust, Stafford, United Kingdom, **6** Department of Social Work and Sociology, University of Zambia, School of Humanities and Social Sciences, Lusaka, Zambia, **7** Department of Social Work, Child Welfare and Social Policy, Oslo Metropolitan University, Oslo, Norway

* tore.bonsaksen@inn.no

**Data Availability Statement:** The data underpinning the conclusions in this study can be found here: https://doi.org/10.18710/B69ATB.

## Abstract

Sleep problems are commonly related to stress and mental health problems. However, social media use has become widespread in the general population during recent years, and their addictive potential may influence people's sleep routine. In addition, the COVID-19 pandemic gave rise to substantial mental health problems in the population, and restrictions in social life gave social media a unique position as means for both entertainment and inter-personal contact. The aim of the study was to examine sleep problems in relationship to social media use in a cross-national sample two years after the COVID-19 outbreak. Participants were 1405 adults from four countries who completed a cross-sectional online survey. The data were analyzed with independent samples *t*-tests, Chi Squared tests of independence, and single and multivariable logistic regression analyses. Of the 858 (61.1%) participants who reported sleep problems during the past weeks, a substantial proportion (*n* = 353, 41.1%) related their sleep problems to their experience with COVID-19. With adjustments for age, gender, employment, and psychological distress, more hours of daily social media use was not significantly associated with sleep problems. However, higher age (OR: 1.13, *p* = 0.01), female gender (OR: 1.69, *p*<0.001), having employment (OR: 1.34, *p* = 0.04), and higher levels of psychological distress (OR: 1.20, *p*<0.001) were independently associated with sleep problems. While the debate about the pros and cons of social media use continue, this study suggests that higher levels of social media use was not of great concern for people's sleep quality in the late COVID-19 pandemic phase. However, other aspects of social media use (eg, time of the day, content of interactions, associated stress experience) may be more relevant for understanding sleep problems and may be taken into consideration for people who experience such problems.

**Funding:** The publication of this article was funded by University of Inland Norway. Otherwise, the funder had no role in the research process.

**Competing interests:** The authors have declared that no competing interests exist.

**Abbreviations:** CI, Confidence interval; COVID-19, Corona Virus Disease, first detected in 2019; GHQ, General Health Questionnaire; OR, Odds ratio.

## Introduction

The use of social media has become widespread during later years, and many–first and foremost younger people–spend considerable time engaging with them [1,2]. The opportunities that were introduced with social media were manifold. It became possible to engage with people and events across the world only by a keystroke, and news, entertainment, and a plethora of special interest groups were instantly accessible. This led to optimism, particularly due to the prospects of being able to utilize such new technologies to support interpersonal connection and wellbeing, as some studies showed [3,4].

Over the years, however, researchers have inquired about a range of possible negative consequences of social media, often including poorer mental health, or factors assumed to be in the pathway between social media use and poorer mental health. Theoretical models focusing on motivation, such as the uses and gratifications theory [5], posit that people use social media for the perceived gratifications they can provide. However, if the person's needs (e.g., for belonging and support) exceed the perceived gratification of these needs, using social media may become excessive based on the hope that more intense use will result in the desired needs gratification [6]. Excessive social media use may develop into social media addiction, generally referring a maladaptive pychological dependency of online social networks to the extent that behavioral addiction symptoms occur [6]. In turn, studies suggest that such behaviors may impede normal functioning in central areas of life such as relationships, studies and work, and health and well-being [7]. With a view to mental health, studies have shown associations between higher social media use and factors such as stress [8,9], anxiety and depression symptoms [10–13], and loneliness [14–16]. In addition, motives for social media use, and not least the personal experience from using them, may be even more relevant for health outcomes than the actual time spent using them, as suggested from recent studies [13,17–19].

Health promoting behaviors, such as physical activity and adequate sleep, have also been studied in relation to social media use. Studies have shown lower levels of physical activity [20], and lower levels of adequate sleep [21,22], among people with higher levels and intensity of social media use. Sleep quality and sleep problems may also be related to the time of the day when social media are used, as late night use has been shown to be related to poorer sleep quality among university students while using them earlier in the evening was related to better sleep [23]. In addition, physical activity and sleep quality have been shown to partially mediate relationships between social media use and mental health and wellbeing [24,25]. In other words, the relationship between social media use and poorer mental health and wellbeing may in part be accounted for by the negative impact of social media use on health-promoting lifestyle behaviors. Considering the potential of social media to interfere with sleep [26,27], the higher psychological distress levels among those with sleep problems [28], and the increasing attention to social media addiction across the world [29], further investigations into the role of social media for sleep problems are warranted.

The pandemic that besieged the world in early 2020, referred to as the COVID-19 pandemic, did not spare any country its devastating impacts [30]. Contracting the disease had multiple negative effects, most notably difficulty breathing, and could be fatal, in particular for old people and people with chronic health problems. In fact, by January 2022 more than six million people worldwide had died from the disease [30]. Although the number of cases and deaths varied substantially between countries, the pandemic had a range of other consequences, such as schools, shops and workplaces closing, leading to a number of challenges for individuals and families such as home schooling, working remotely, or even losing one's job. Thus, many people's daily routines changed radically during the pandemic and a general sense of crisis emerged, with inherent risks of inducing sleep problems. Indeed, several studies

surfaced that focused on sleep problems during the pandemic. A number of systematic reviews and meta-analyses of such studies reported a global prevalence of sleep problems in the general population. For example, Alimoradi and co-workers reported a prevalence of 18% [31], whereas Jahrami and co-workers reported a prevalence of 32% [32]. Despite diverging prevalence rates across these meta-analyses, they substantiate that sleep problems was a major public health issue at the time. Moreover, while the psychological consequences of the pandemic were widespread in the population, those who had personal experience with the disease, or had experienced that someone in the family had been ill, were reported to have comparably poorer mental health and more worry about themselves and their close ones [33], factors which in turn have been closely linked with sleep problems [34].

Due to restrictions during the pandemic, individuals leaned more on social media and other electronic methods of communication with family and friends, but also more generally has the use of social media increased over the past years. Between 2015 and 2023 the average daily time spent using social media increased by approximately 40 minutes, with the recent average being reported at 151 minutes [35]. Considering the evidence, the increased use of social media may give rise to more sleep problems among people. However, while the literature on associations between social media use, sleep problems, and poorer mental health is substantial, many of these studies have focused on children and adolescents [e.g., 26,27,36,37]. Moreover, these associations may be different in a pandemic context where a range of other factors and events can give cause for sleep problems. The gap in the literature concerning these associations among the general adult population, and during the extraordinary context of the COVID-19 pandemic, constitutes the rationale for this study. The aim of the study was to answer the question: What is the association between daily hours of social media use and self-reported sleep problems in a cross-national adult population in the late COVID-19 pandemic phase?

## Materials and methods

### Design

The study had a cross-sectional survey design. It study was conducted in four countries (Norway, UK, USA, and Australia) during the same period (20. November 2021–31. January 2022), which was approximately two years after the COVID-19 pandemic outbreak. At the time of the data collection, the fifth wave of COVID-19 virus transmission had emerged. While there were regional variations in restrictions within the four countries, all countries required use of face masks in public for parts of the period, indicating that coping with pandemic restrictions was still an aspect of people's everyday lives. Information about the study was disseminated through several social media channels, including Facebook, Twitter, and LinkedIn, and postings on these social media included a direct link to the survey website. Project group members in each of the four countries were responsible for conducting the survey and collecting the data. The online survey was open for participation among the general adult population (> 18 years of age) living in the four countries.

### Sample

The survey relied on convenience sampling by self-selection and recruited a total number of 1649 participants. The invitation to participate stated that all adults over the age of 18 years were eligible for participation, provided that they were able to understand the language in which the survey was distributed (ie, Norwegian in Norway, and English in USA, UK, and Australia). Participants with missing values on one or more of the included variables in the study ($n = 244$, 14.8%) were removed from the sample prior to analysis, resulting in a sample

of 1405 participants included in the analyses. The participants were from Norway ($n = 218$, 15.5%), UK ($n = 220$, 15.7%), USA ($n = 761$, 54.2%), and Australia ($n = 206$, 14.7%). There was a larger proportion of females than males ($n = 1105$, 78.6% versus $n = 300$, 21.4%).

## Measures

**Sleep problems.** Sleep problems was assessed with the following question: "In the past weeks, do you have difficulty sleeping?" Further, attribution of sleep problems to COVID-19 was assessed with the question: "If you have difficulty sleeping, do you think that it is related to your experience with the COVID-19 pandemic?" Both questions had response options 'yes' and 'no'.

**Daily social media use.** The participants indicated the amount of time they had spent on social media during a typical day during the last month. In line with Ellison and colleagues [38], response options were less than 10 minutes, 10–30 minutes, 31–60 minutes, 1–2 hours, 2–3 hours, and more than three hours. Higher values on the social media use variable indicated more daily time spent using social media.

**Sociodemographic characteristics.** Sociodemographic variables included country (Norway, UK, USA, Australia), age group (18–29, 30–39, 40–49, 50–59, 60–69, 70 years and above), gender (male versus female), employment status (full-time or part-time employment versus not employed), and education level (higher [ie, BSc degree or higher] versus lower level).

**COVID-19 infection experience.** Infection experience was assessed with the question: "Have you been infected with COVID-19?" Response options were 'yes' and 'no'.

**Psychological distress.** Psychological distress was assessed with the *General Health Questionnaire-12* (GHQ-12), which is a widely used self-report measure [39–41]. A large number of studies in a variety of populations, including general adult, clinical, work- and student populations, have suggested that it is a valid measure that can be used across samples and contexts [40,42–46]. It has been translated from English to several other language, among these Norwegian [47], and validity has been established also for the Norwegian version.[48] Of the GHQ items, six are positively phrased (eg, 'able to enjoy day-to-day activities') and six are negatively phrased (eg, 'felt constantly under strain'). The study participants responded by indicating the degree to which they had experienced the relevant item content during the two past weeks: 'less than usual (0), 'as usual' (1), 'more than usual' (2) or 'much more than usual' (3). Positively formulated items were recoded prior to analysis, so that higher scores indicated more psychological distress (possible score range 0–36). Cronbach's α was 0.90 in the current sample.

## Data analysis

All variables used in the study were analyzed descriptively, with frequencies and percentages for categorical variables and means and standard deviations for continuous variables. Comparisons between participants with and without sleep problems were performed with Chi Squared tests of independence (categorical variables) and independent samples *t*-tests (continuous variables). Single and multivariable logistic regression analyses were used to examine associations between independent variables (age, gender, employment, education level, COVID-19 infection experience, psychological distress, and daily time spent on social media) and sleep problems. To avoid suppressor effects, independent variables were carried over from the single variable analysis to the multivariable analysis in the case of associations showing p≤0.30 in the single variable analysis [49]. In the multivariable analysis, all independent variables were entered in one block. Effect sizes are reported as odds ratio (OR) along with their corresponding 95% confidence intervals (CI). Statistical significance was set at $p < 0.05$.

## Ethics

The study was performed in line with the principles of the Declaration of Helsinki, and the researchers adhered to all relevant regulations in their respective countries concerning ethics and data protection. Ethical approvals and permission to conduct the study were received from the following review boards: OsloMet (20/03676) and the Regional Committees for Medical and Health Research Ethics (REK; ref. 132066, 6. April 2020) in Norway, by the University of Michigan Institutional Review Board for Health Sciences and Behavioral Sciences (IRB HSBS) and designated as exempt (HUM00180296, 23. April 2020) in USA, University of Central Lancashire (Health Ethics Review Pane; HEALTH 0246, 12. November 2021) in the UK, and by the University of Queensland (2021/HE002544, 16. November 2021) in Australia. All data collected in this study were anonymous. The participants gave their written consent to participate as part of their response to the survey.

## Results

### Sample characteristics

The age composition of the sample differed significantly between countries, with more participants in the younger age groups coming from the USA and UK, and with more participants in the oldest age groups coming from Australia. The gender distributions were similar between the four countries. Compared to the other countries, Australia had lower proportions employed, lower proportions with higher education, and lower proportions living with a spouse or partner. Australia also had the lowest proportion with COVID-19 infection experience, while the proportions in the USA and UK were substantially higher than in Australia and Norway.

### Sample characteristics in relation to sleep problems

Of the 858 (61.1%) participants who reported sleep problems during the past weeks, a substantial proportion ($n$ = 353, 41.1%) related their sleep problems to their experience with COVID-19. Women reported sleep problems more often than men (64% versus 49%, $p<0.001$). Compared to their counterparts, participants with sleep problems had higher mean GHQ rating (16.9 versus 11.7, $p<0.001$), indicating higher levels of psychological distress, and reported more time spent on social media (mean rating 4.4 versus 4.1, $p<0.001$). No other variables showed a significant relationship to sleep problems in the initial analyses. A detailed overview of the sample characteristics as related to sleep problems is provided in Table 1.

### Adjusted associations with sleep problems

Based on the single variable logistic regression analyses, female gender, higher GHQ ratings, and more time spent on social media were significantly associated with sleep problems (all $p<0.001$). Age and employment were carried over to the multivariable analysis due to associations with sleep problems showing $p \leq 0.30$.

In the multivariable (adjusted) analysis, sleep problems were associated with higher age (OR: 1.13, $p$ = 0.01), female gender (OR: 1.69, $p<0.001$), having employment (OR: 1.34, $p$ = 0.04), and higher GHQ ratings (OR: 1.20, $p<0.001$). Time spent on social media was no longer significantly associated with sleep problems after adjustment (OR: 1.06, $ns$). Results from the single and multivariable logistic regression analyses are displayed in Table 2.

### Post-hoc moderation analysis

In a final step, we examined whether gender moderated the association between social media use and sleep problems. Predictors were included as in the previous multivariable logistic

**Table 1. Sample characteristics by sleep problems.**

| Characteristics | All n = 1405 | Sleep problems n = 858 (61.1%) | No sleep problems n = 547 (38.9%) | p |
|---|---|---|---|---|
| *Country* | | | | |
| Norway | 218 | 134 (61.5) | 84 (38.5) | 0.09 |
| USA | 761 | 445 (58.5) | 316 (41.5) | |
| UK | 220 | 149 (67.7) | 71 (32.3) | |
| Australia | 206 | 130 (63.1) | 76 (36.9) | |
| *Age group n (%)* | | | | |
| 18–29 | 226 | 133 (58.8) | 93 (41.2) | 0.35 |
| 30–39 | 362 | 232 (64.1) | 130 (35.9) | |
| 40–49 | 394 | 242 (61.4) | 152 (38.6) | |
| 50–59 | 223 | 141 (63.2) | 82 (36.8) | |
| 60–69 | 134 | 75 (56.0) | 59 (44.0) | |
| 70 and above | 66 | 35 (53.0) | 31 (47.0) | |
| *Gender n (%)* | | | | |
| Male | 300 | 146 (48.7) | 154 (51.3) | <0.001 |
| Female | 1105 | 712 (64.4) | 393 (35.6) | |
| *Employment n (%)* | | | | |
| Yes (full-time or part-time) | 1020 | 633 (62.1) | 387 (37.9) | 0.22 |
| No | 385 | 225 (58.4) | 160 (41.6) | |
| *Education level n (%)* | | | | |
| Higher education | 1067 | 656 (61.5) | 411 (38.5) | 0.57 |
| Lower education | 338 | 202 (59.8) | 136 (40.2) | |
| *Experienced COVID infection n (%)* | | | | |
| No infection | 1143 | 703 (61.5) | 440 (38.5) | 0.48 |
| Infection | 262 | 155 (59.2) | 107 (40.8) | |
| *GHQ scores M (SD)* | 1405 | 16.9 (6.2) | 11.7 (4.8) | <0.001 |
| *Time on social media M (SD)* | 1405 | 4.4 (1.3) | 4.1 (1.4) | <0.001 |

Note. *p*-values are from Chi Square tests of independence (categorical variables) and independent samples *t*-test (countinuous variables).

regression analysis, but the interaction term social media use × gender was included in a second block of the analysis. The analysis showed that the interaction effect was not statistically significant (OR: 1.01, *ns*), indicating that the association between social media use and sleep problems was uniform (i.e., non-existing) for men and women.

## Discussion

Although social media has often been blamed for disrupting the sleep and subsequent psychosocial wellbeing of individuals [26,27], the findings from this study suggest that in the pandemic context, they may be falsely blamed. The association between time spent on social media and sleep problems was no longer statistically significant in the multivariable analysis adjusting for other variables. This suggests that while there was an initial bivariate relationship between social media usage and sleep problems, this relationship was likely the result of one or more variables being related to both social media use and sleep problems (confounding). Female gender, higher psychological distress, higher age, and having employment were significantly associated with sleep problems after adjustment.

Noteworthy associations with sleep problems were shown for female gender and psychological distress, and there is reason to suggest that these factors might also explain the results for

**Table 2. Unadjusted and adjusted logistic regression analyses showing associations with sleep problems ($n$ = 1405).**

| Independent variables | Unadjusted analysis | | | Adjusted analysis | | |
|---|---|---|---|---|---|---|
| | OR | 95% CI | *p* | OR | 95% CI | *p* |
| *Age group* | 0.96 | 0.89–1.04 | 0.30 | 1.13 | 1.03–1.24 | 0.01 |
| *Gender* | | | | | | |
| Male | | reference | | | reference | |
| Female | 1.91 | 1.48–2.47 | <0.001 | 1.69 | 1.27–2.25 | <0.001 |
| *Employment* | | | | | | |
| No | | reference | | | reference | |
| Yes (full-time or part-time) | 1.16 | 0.92–1.48 | 0.22 | 1.34 | 1.02–1.77 | 0.04 |
| *Education level* | | | | | | |
| Lower level education | | reference | | | | |
| Higher level education | 1.08 | 0.84–1.38 | 0.57 | | | |
| *Experienced COVID infection* | | | | | | |
| No infection | | reference | | | | |
| Infection | 0.91 | 0.69–1.19 | 0.48 | | | |
| *GHQ scores* | 1.20 | 1.17–1.23 | <0.001 | 1.20 | 1.17–1.23 | <0.001 |
| *Time on social media* | 1.17 | 1.08–1.27 | <0.001 | 1.06 | 0.96–1.16 | 0.25 |

Note. On continuous variables, OR refers to the change in likelihood for having sleep problems for each unit increase in the predictor variable. Multivariable model: Chi Square p<0.001, Nagelkerke $R^2$ = 0.255.

social media use. The higher levels of psychological distress and sleep problems among women has been well documented, both prior to and during the COVID-19 pandemic [50–52]. Studies have also pointed to girls and women as more frequent users of social media, compared to boys and men [53–55], and also that the association between higher social media use and lower wellbeing is stronger for girls [54]. Thus, if adult women use social media more than men and also respond to their social media experiences with more distress, this may explain why the association between social media use and sleep problems was weakened and no longer statistically significant after controlling for gender and psychological distress.

While female gender and higher psychological distress are linked, the multivariate model showed that both of these variables had a unique relationship with sleep problems. Thus, more sleep problems among women than men may partially, but not fully, be explained with reference to higher psychological distress levels among women. The processes driving sleep problems may therefore concern a range of other factors than psychological distress. Previous research has suggested biological factors such as differences in sex steroids [56] and women's menstruational cycle [57], but women may also be psychologically more prone to lie awake pondering about things, without necessarily feeling distressed. As we do not have data to explore these possibilities further, we suggest that future research may address the processes driving sleep problems in a gender perspective.

The association between age and sleep problems was suppressed in the unadjusted analysis, but became statistically significant when cancelling out the effects of the other included variables. During the pandemic crisis, people of younger age have reported higher levels of psychological distress, compared to older people [58,59], despite older people being more prone to become seriously ill from contracting COVID. Thus, confounding from psychological distress may explain the initial results for the association between age and sleep problems. However, after adjustment, higher age was associated with sleep problems, which is in line with results from previous studies demonstrating that sleep problems is common among people of older age [60,61].

Having employment was significantly associated with sleep problems in the multivariable analysis. This was a surprise, considering that not having a job would commonly be related to more economic concerns, which in turn may affect sleep negatively [62–64]. However, having a job may also be the cause of daily concerns, both directly related to the work tasks and the work environment [65], but also indirectly, considering for example commuting and balancing work-life commitments. Moreover, being employed in the COVID-19 era may be particularly stressful, given the demands for adapting to new ways of organizing work, including more digitalization, more working from home–and therefore also more role overload and family distraction [66]. Possibly, these and other aspects of having employment during the COVID-19 pandemic might translate into more sleep problems.

No association was shown between having had the COVID disease and sleep problems. While previous studies have shown considerably more sleep problems among COVID patients compared to the general population (75% vs. 32%) [32] one should consider the risks as perceived in different phases of the pandemic. In the early phase, the number of cases and deaths was rocketing and there were still no vaccines available, both of which contributed to incrasing people's psychological distress [67]. In later phases of the pandemic, people had had time to adapt to the situation, vaccines had become available, and many had already been infected with COVID and had recovered. All of these factors would indicate that at the time of the data collection for this study, approximately two years after the pandemic outbreak, sleep problems would not be significantly different for people who had been infected with COVID compared to those who had not.

## Limitations

The study had a cross-sectional design, thus the results cannot be used to imply cause and effect relationships. We do not have information about the specific social media people used, and we do not know how, why, or when they were used, since this study only collected data on time spent using social media. As the study is concerned with associations with sleep problems, the lacking information about these aspects of social media use represent a limitation. The recruitment strategy, relying on dissemination on social media, required that all participants had to use a computer, tablet, or smartphone to be able to access and respond to the survey. Thus, the population should be specified as 'users of social media' as a subset of the general population. In the dataset we noticed more participants in the younger age groups and assume this may have been a result of the data collection strategy. The study may have limited transferability to the general population, given the skewed sample distributions on age, gender and education. Moreover, while all countries are considered representative of Western culture, there may still be cultural differences between the countries. Given that larger proportions of younger participants came from the USA and UK, and larger proportions of older participants came from Australia, it is possible that the age-related results are influenced by culture. Limitations also include that the results are solely based on the participants' self-report, and the use of measurement methods–including the sleep problems measure–being suboptimal and with unknown psychometric properties. Suggestions for the methodological improvement of future studies in this area thereore include longitudinal data collections, sampling methods that secure the sample's representativity of the selected population, use of well-established and good quality measurements, and a more detailed inquiry into the various aspects of social media use that goes beyond time use.

## Conclusions

To improve the health and wellness of individuals globally, knowledge about their influencing factors are vital. This study showed, in contrast to expectations, that time spent using social

media was unrelated to people's sleep problems during the late phase of the COVID-19 pandemic. Higher psychological distress levels, on the other hand, were associated with sleep problems, and women were more inclined to report sleep problems than men. While the results suggest that time on social media in and of itself may not play a role for people's sleep quality, other aspects of social media use (eg, time of the day, content of interactions, associated stress experience) may be more relevant for understanding sleep problems. Investigating the complexities of social media use in relation to sleep patterns and problems appears to be a fruitful way forward. In addition, as the study showed that characteristics such as age, gender, psychological distress, and employment contribute to predicting sleep disturbances, it underscores that these factors should continue to be used for adjustment in the future exploration of other predictors. Increasing the understanding of the complexity of factors related to sleep problems can support the development of prevention and intervention strategies to improve sleep and overall psychosocial wellness.

## Supporting information

**S1 File. Ethics approval for Norway.**
(PDF)

**S2 File. Ethics approval for USA.**
(PDF)

**S3 File. Ethics approval for Australia.**
(PDF)

**S4 File. Ethics approval for UK.**
(PDF)

**S5 File. Translated ethics approval for Norway.**
(PDF)

**S6 File. OsloMet university approval Norway.**
(PDF)

## Acknowledgments

The authors are grateful for the time and efforts made by the survey participants, and thank Hilde Thygesen (University of South-Eastern Norway, Norway), Mary Ruffolo (University of Michigan, USA) and Janni Leung (The University of Queensland, Australia) for their contributions to the research group in general and for their contributions to the data collection in particular.

## Author Contributions

**Conceptualization:** Tore Bonsaksen, Daicia Price, Gary Lamph, Isaac Kabelenga, Amy Østertun Geirdal.

**Data curation:** Tore Bonsaksen, Amy Østertun Geirdal.

**Formal analysis:** Tore Bonsaksen.

**Funding acquisition:** Tore Bonsaksen.

**Investigation:** Tore Bonsaksen, Daicia Price, Gary Lamph, Isaac Kabelenga, Amy Østertun Geirdal.

**Methodology:** Tore Bonsaksen, Daicia Price, Gary Lamph, Isaac Kabelenga, Amy Østertun Geirdal.

**Project administration:** Amy Østertun Geirdal.

**Supervision:** Amy Østertun Geirdal.

**Validation:** Tore Bonsaksen.

**Visualization:** Tore Bonsaksen.

**Writing – original draft:** Tore Bonsaksen.

**Writing – review & editing:** Tore Bonsaksen, Daicia Price, Gary Lamph, Isaac Kabelenga, Amy Østertun Geirdal.

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
