## [Decision Letter · Decision Letter 0]

26 Dec 2024

PONE-D-24-45013Sleep problems were unrelated to social media use in the late COVID-19 pandemic phase: a cross-national studyPLOS ONE

Dear Dr. Bonsaksen,

Thank you for submitting your manuscript to PLOS ONE. After careful consideration, we feel that it has merit but does not fully meet PLOS ONE’s publication criteria as it currently stands. Therefore, we invite you to submit a revised version of the manuscript that addresses the points raised during the review process.

We look forward to receiving your revised manuscript.

Kind regards,

Javier Fagundo-Rivera, PhD

Academic Editor

PLOS ONE

Journal Requirements:

2. Please include a complete copy of PLOS’ questionnaire on inclusivity in global research in your revised manuscript. Our policy for research in this area aims to improve transparency in the reporting of research performed outside of researchers’ own country or community. The policy applies to researchers who have travelled to a different country to conduct research, research with Indigenous populations or their lands, and research on cultural artefacts. The questionnaire can also be requested at the journal’s discretion for any other submissions, even if these conditions are not met.  

Please find more information on the policy and a link to download a blank copy of the questionnaire here: https://journals.plos.org/plosone/s/best-practices-in-research-reporting. 

Please upload a completed version of your questionnaire as Supporting Information when you resubmit your manuscript.

4. Please note that funding information should not appear in the Acknowledgments section or other areas of your manuscript. We will only publish funding information present in the Funding Statement section of the online submission form. Please remove any funding-related text from the manuscript. 

5. We note that you have indicated that there are restrictions to data sharing for this study. For studies involving human research participant data or other sensitive data, we encourage authors to share de-identified or anonymized data. However, when data cannot be publicly shared for ethical reasons, we allow authors to make their data sets available upon request. For information on unacceptable data access restrictions, please see http://journals.plos.org/plosone/s/data-availability#loc-unacceptable-data-access-restrictions. 

6. In this instance it seems there may be acceptable restrictions in place that prevent the public sharing of your minimal data. However, in line with our goal of ensuring long-term data availability to all interested researchers, PLOS’ Data Policy states that authors cannot be the sole named individuals responsible for ensuring data access (http://journals.plos.org/plosone/s/data-availability#loc-acceptable-data-sharing-methods).

7.  Your ethics statement should only appear in the Methods section of your manuscript. If your ethics statement is written in any section besides the Methods, please delete it from any other section. 

**Additional Editor Comments:**

Dear authors,

Three reviewers have evaluated your investigation.

Please, respond to all their comments so we can reach a final decision after carefully considering your responses.

Thank you.

Reviewers' comments:

Reviewer's Responses to Questions

**Comments to the Author**

1. Is the manuscript technically sound, and do the data support the conclusions?

Reviewer #1: Yes

Reviewer #2: Partly

Reviewer #3: Yes

2. Has the statistical analysis been performed appropriately and rigorously? 

Reviewer #1: Yes

Reviewer #2: No

Reviewer #3: Yes

3. Have the authors made all data underlying the findings in their manuscript fully available?

Reviewer #1: Yes

Reviewer #2: No

Reviewer #3: No

4. Is the manuscript presented in an intelligible fashion and written in standard English?

Reviewer #1: Yes

Reviewer #2: Yes

Reviewer #3: Yes

5. Review Comments to the Author

Reviewer #1: The article stands out for its quality of exposition. The analysis of the questionnaire is conducted rigorously and contributes to developing a homogeneous and coherent argument in line with the research question formulated in the paper. However, I would suggest emphasizing the gender differences in the text, as the sample shows a significant imbalance in this area. Additionally, I would recommend updating the literature section with more recent references and including a brief paragraph that informs the reader about the current line of research exploring the link between social media addictions (such as Instagram, TikTok, and WhatsApp) and difficulties in falling asleep.

Reviewer #2: This research explored the association between social media use and sleep problems approximately two years after the outbreak of the COVID-19 pandemic through a web-based survey.

However, the significance of this research is unclear due to the limited information provided about the main variables, namely sleep problems and social media use. Below, I have outlined specific concerns for each section.

METHODS

The authors should explain the social circumstances (e.g., restrictions) in each country where the survey was conducted during the research period. Additionally, the appropriateness of the survey period (November–January) should be considered, as social media use might increase, and physical activity might decrease during the winter season (except in Australia).

Sampling bias might be present, as the participants were recruited through social media. This could mean the study sample was limited to active social media users.

The validity of the sleep problem question is questionable. The question provides only limited information about sleep issues, despite there being many types of sleep problems (e.g., insomnia, delayed sleep phase, sleep deficit). Similarly, the question on social media use also appears limited in scope. The authors mentioned in the introduction that motives for social media use may be more relevant to health outcomes than the actual time spent using them, but no data on these motives were collected.

RESULTS

The means of time on social media (4.4 and 4.1) is unclear. Does it mean 4.4 hours? Given the response option, it is not thought to be continuous value. If so, calculating average is not appropriate.

The data were collected from four countries, where the seasons and pandemic circumstances are different between the countries. This factor should be considered for analysis.

Reviewer #3: * General Aspects

We appreciate the opportunity to review this article because it is, in fact, a very relevant topic. Regarding originality, it can be considered that the article does not make a particularly relevant contribution in this field of study, although it proposes to analyze the relationship between the use of social networks, sleep problems and mental health in the adult population in general, in the context of the covid-19 pandemic.

The manuscript is written in a clear and coherent way, and the theme is very important, especially if we consider that technology and social networks are part of our days and can greatly affect our well-being and quality of life.

Therefore, here are some aspects that may strengthen the arguments presented and improve the article.

*Title:

It is consistent with the content presented, clear and objective. It arouses the reader's interest, because it is a cross-national study.

*Summary:

The abstract is clear and well organized, following the structure of the article. The introduction can, however, be improved by deepening the relationship between the use of social networks, sleep problems and mental health in the adult population in a pandemic context. It may also be useful to include more clearly the contribution/implications of the study for research in this area.

*Introduction:

The introduction adequately presents the context and importance of the study. It is clearly worded, but perhaps it is a little shorter than desirable in terms of existing research on (1) people's motivations for using social media and (2) personal experience of using it. As the authors point out, these can be very relevant aspects for mental health outcomes (even more than the time spent using social networks).

It is important that the introduction situates the existing knowledge on the topic under study, helping to formulate the hypotheses/arguments that will be developed in the article. There is also a need to make a link between the previous work and this article, so it would be interesting to explore further why the approach adopted is innovative. In other words, it could be useful to clarify the originality and relevance of the study, since the innovative nature of the study is not entirely clear.

It would also be appropriate for the study to have a theoretical perspective that would allow a reading and interpretation of the results obtained. The study seems atheoretical.

The introduction ends with the presentation of the purpose of the study – which is very appropriate.

*Methodology:

The authors adequately describe how they obtained the data, and the methodology used to analyze them. There are some reservations about the measures used to assess the daily use of social networks (using only one question to assess the amount of time people spent on social networks during a typical day during the last month can be reductive), not least because it would have been very interesting to also evaluate people's motivations for using social networks and personal experience of using it. In addition, I did not understand exactly the purpose of questioning whether the person was infected with covid-19.

Finally, from a methodological point of view, the fact that data collection took place approximately two years after the pandemic outbreak may have greatly affected the results obtained.

From the point of view of the characteristics of the participants, and in the case of a cross-national study, it may make sense to present a brief characterization of the participants by country as this can be useful to understand the results obtained. As regards the sample size, it is considered an acceptable size.

*Results

The results section is interesting and focuses on the essentials. However, the results obtained do not seem to add anything very relevant to research in this field. In addition, the pandemic context seems not to have been considered or had no effect on the results obtained.

The results of the logistic regression show that time spent on social networks is not statistically associated with sleep problems, nor are there differences between men and women. That said, the question that arises is whether the results obtained are relevant to the advancement of knowledge in the field?

*Discussion

The discussion begins with a synthesis of the main results, which is appropriate. There is room, however, for a deeper discussion about the results obtained because some arguments would make more sense in the introduction of the article and not in the discussion. At this point, one would expect a more in-depth critical analysis of the main results of the study.

It is suggested, therefore, a greater robustness of the theoretical and empirical framework in the introduction so that in the discussion it is possible to adopt a more analytical and critical lens on the results obtained.

Regarding the main limitations of the study, the composition of the sample may be a weakness (the fact that it includes mainly people of young age and more educated) and this has, as mentioned by the authors, implications for the generalization of the results. Likewise, it would have been useful to use more robust and complete data collection procedures/tools to explore other possibilities in data analysis.

It could also be useful to analyze cultural influence as this is a cross-national study.

It is also essential to make a clear statement about the contribution and/or implications of the study in relation to existing knowledge. What are, in fact, the main conclusions of the study and what implications do they have for the advancement of knowledge in this field?

*Conclusion

Based on the results of the study, it is concluded that the time spent on social media was not related to people's sleep problems during the final phase of the COVID-19 pandemic. Considering that this result does not meet expectations (as mentioned by the authors), it is important to deepen the study and analyze the complexity associated with the use of social networks (e.g., time of day, content of interactions, associated stress experience) regarding the sleep patterns and problems of adults.

6. PLOS authors have the option to publish the peer review history of their article (what does this mean?). If published, this will include your full peer review and any attached files.

Reviewer #1: No

Reviewer #2: No

Reviewer #3: No

---

## [Author Response · Author response to Decision Letter 0]

16 Jan 2025

Authors: Thank you for your comments and suggestions, which we believe have contributed to strengthening the article. The revised manuscript shows all changes tracked. Each comment has been responded to point by point in this letter. We look forward to hearing from you again.

Editor: Dear Dr. Bonsaksen, thank you for submitting your manuscript to PLOS ONE. After careful consideration, we feel that it has merit but does not fully meet PLOS ONE’s publication criteria as it currently stands. Therefore, we invite you to submit a revised version of the manuscript that addresses the points raised during the review process.

Authors: Thank you.

Editor: Please submit your revised manuscript by Feb 09 2025 11:59PM. If you will need more time than this to complete your revisions, please reply to this message or contact the journal office at plosone@plos.org. Please include the following items when submitting your revised manuscript:

Authors: This letter is the rebuttal letter.

Editor:

Authors: Both files have been labeled according to this requirement.

Editor: Authors: We have included our financial disclosure in the cover letter. No figure files have been used.

Editor: If applicable, we recommend that you deposit your laboratory protocols in protocols.io to enhance the reproducibility of your results. Protocols.io assigns your protocol its own identifier (DOI) so that it can be cited independently in the future. For instructions see: https://journals.plos.org/plosone/s/submission-guidelines#loc-laboratory-protocols. Additionally, PLOS ONE offers an option for publishing peer-reviewed Lab Protocol articles, which describe protocols hosted on protocols.io. Read more information on sharing protocols at https://plos.org/protocols?utm_medium=editorial-email&utm_source=authorletters&utm_campaign=protocols.

Authors: No laboratory protocols have been used.

Editor: Journal Requirements: When submitting your revision, we need you to address these additional requirements.

Authors: We have consulted the relevant documents and have modified the files accordingly.

Editor: 2. Please include a complete copy of PLOS’ questionnaire on inclusivity in global research in your revised manuscript. Our policy for research in this area aims to improve transparency in the reporting of research performed outside of researchers’ own country or community. The policy applies to researchers who have travelled to a different country to conduct research, research with Indigenous populations or their lands, and research on cultural artefacts. The questionnaire can also be requested at the journal’s discretion for any other submissions, even if these conditions are not met. 

Please find more information on the policy and a link to download a blank copy of the questionnaire here: https://journals.plos.org/plosone/s/best-practices-in-research-reporting. 

Please upload a completed version of your questionnaire as Supporting Information when you resubmit your manuscript.

Authors: As the policy applies to researchers who have travelled to a different country to conduct their research, it does not apply to this manuscript. The research group (including those acknowledged) represent all of the countries where data collection took place in this study.

Editor: 3. We note that the grant information you provided in the ‘Funding Information’ and ‘Financial Disclosure’ sections do not match. 

Authors: We would like the statement to read as follows: “Funding to cover the APC for the article was granted by University of Inland Norway. Otherwise, no funding was obtained for this study.” However, note two issues: There is no grant number for the financial support we received to cover the APC for this study. In addition, my university recently changed its name from Inland Norway University of Applied Sciences to University of Inland Norway. The latter is not yet possible to select from the list of funders, thus, discrepancies may still be found related to statements of funding.

Editor: 4. Please note that funding information should not appear in the Acknowledgments section or other areas of your manuscript. We will only publish funding information present in the Funding Statement section of the online submission form. Please remove any funding-related text from the manuscript. 

Authors: Removed as required. 

Editor: 5. We note that you have indicated that there are restrictions to data sharing for this study. For studies involving human research participant data or other sensitive data, we encourage authors to share de-identified or anonymized data. However, when data cannot be publicly shared for ethical reasons, we allow authors to make their data sets available upon request. For information on unacceptable data access restrictions, please see http://journals.plos.org/plosone/s/data-availability#loc-unacceptable-data-access-restrictions. 

Authors: Please refer to our response to Editor comment no. 6.

Editor: Before we proceed with your manuscript, please address the following prompts:

6. In this instance it seems there may be acceptable restrictions in place that prevent the public sharing of your minimal data. However, in line with our goal of ensuring long-term data availability to all interested researchers, PLOS’ Data Policy states that authors cannot be the sole named individuals responsible for ensuring data access (http://journals.plos.org/plosone/s/data-availability#loc-acceptable-data-sharing-methods).

Authors: We have modified our position on this matter, and we have provided access to the minimal dataset allowing for the replication of the analyses performed in the study. The data can be accessed from here: https://dataverse.no/dataset.xhtml?persistentId=doi:10.18710/B69ATB. See also our modified data availability statement in the manuscript.

Editor: 7. Your ethics statement should only appear in the Methods section of your manuscript. If your ethics statement is written in any section besides the Methods, please delete it from any other section. 

Authors: Removed from all other sections.

Editor: 8. Please include captions for your Supporting Information files at the end of your manuscript, and update any in-text citations to match accordingly. Please see our Supporting Information guidelines for more information: http://journals.plos.org/plosone/s/supporting-information. 

Authors: Captions for supplementary files have been included at the end of the manuscript, and the uploaded supplementary files have been labeled according to the guidelines.

Editor: Dear authors, Three reviewers have evaluated your investigation. Please, respond to all their comments so we can reach a final decision after carefully considering your responses.

Authors: All comments have been responded to in this letter.

Reviewers:

1. Is the manuscript technically sound, and do the data support the conclusions?

Reviewer #1: Yes

Reviewer #2: Partly

Reviewer #3: Yes

Authors: Please refer to our responses to the specific reviewer comments.

2. Has the statistical analysis been performed appropriately and rigorously? 

Reviewer #1: Yes

Reviewer #2: No

Reviewer #3: Yes

Authors: Please refer to our responses to the specific reviewer comments.

3. Have the authors made all data underlying the findings in their manuscript fully available?

Reviewer #1: Yes

Reviewer #2: No

Reviewer #3: No

Authors: Please refer to our responses to the specific reviewer comments. Also, note that we have deposited the minimal dataset to a public repository.

4. Is the manuscript presented in an intelligible fashion and written in standard English?

Reviewer #1: Yes

Reviewer #2: Yes

Reviewer #3: Yes

5. Review Comments to the Author

Reviewer #1: The article stands out for its quality of exposition. The analysis of the questionnaire is conducted rigorously and contributes to developing a homogeneous and coherent argument in line with the research question formulated in the paper. However, I would suggest emphasizing the gender differences in the text, as the sample shows a significant imbalance in this area. 

Authors: We agree that the gender differences are important, and we would like to refer to the Discussion section where two full paragraphs are devoted to this issue (note that the numbered references below have changed when copying the text into the new document): 

“Noteworthy associations with sleep problems were shown for female gender and psychological distress, and there is reason to suggest that these factors might also explain the results for social media use. The higher levels of psychological distress and sleep problems among women has been well documented, both prior to and during the COVID-19 pandemic [1-3]. Studies have also pointed to girls and women as more frequent users of social media, compared to boys and men [4-6], and also that the association between higher social media use and lower wellbeing is stronger for girls [5]. Thus, if adult women use social media more than men and also respond to their social media experiences with more distress, this may explain why the association between social media use and sleep problems was weakened and no longer statistically significant after controlling for gender and psychological distress. 

While female gender and higher psychological distress are linked, the multivariate model showed that both of these variables had a unique relationship with sleep problems. Thus, more sleep problems among women than men may partially, but not fully, be explained with reference to higher psychological distress levels among women. The processes driving sleep problems may therefore concern a range of other factors than psychological distress. Previous research has suggested biological factors such as differences in sex steroids [7] and women’s menstruational cycle [8], but women may also be psychologically more prone to lie awake pondering about things, without necessarily feeling distressed. As we do not have data to explore these possibilities further, we suggest that future research may address the processes driving sleep problems in a gender perspective.”

R1: Additionally, I would recommend updating the literature section with more recent references and including a brief paragraph that informs the reader about the current line of research exploring the link between social media addictions (such as Instagram, TikTok, and WhatsApp) and difficulties in falling asleep.

Authors: The introduction section has been updated with more materials and references concerning social media addiction and sleep problems.

Reviewer #2: This research explored the association between social media use and sleep problems approximately two years after the outbreak of the COVID-19 pandemic through a web-based survey. However, the significance of this research is unclear due to the limited information provided about the main variables, namely sleep problems and social media use. Below, I have outlined specific concerns for each section.

Authors: Please see our responses to the specific concerns listed below.

R2: METHODS

The authors should explain the social circumstances (e.g., restrictions) in each country where the survey was conducted during the research period. Additionally, the appropr

---

## [Editor Report · Decision Letter 1]

17 Jan 2025

Sleep problems were unrelated to social media use in the late COVID-19 pandemic phase: a cross-national study

PONE-D-24-45013R1

Dear Dr. Bonsaksen,

We’re pleased to inform you that your manuscript has been judged scientifically suitable for publication and will be formally accepted for publication once it meets all outstanding technical requirements.

Kind regards,

Javier Fagundo-Rivera, PhD

Academic Editor

PLOS ONE

Additional Editor Comments (optional):

Dear Authors,

Thank you for your responses and comments to the Reviewers.

Suggestions have been considered and all the comments have been discussed successfully.

Given my opinion and that of the Reviewers, this manuscript could be accepted.

---

## [Editor Report · Acceptance letter]

21 Jan 2025

PONE-D-24-45013R1 

PLOS ONE

Dear Dr. Bonsaksen, 

I'm pleased to inform you that your manuscript has been deemed suitable for publication in PLOS ONE. Congratulations! Your manuscript is now being handed over to our production team.

Kind regards, 

on behalf of

Dr. Javier Fagundo-Rivera 

Academic Editor

PLOS ONE
